

# Mesoscale spatiotemporal variability in a complex host-parasite system influenced by intermediate host body size

Sara M. Rodríguez[1,2,*] and Nelson Valdivia[2,3,*]

[1] Programa de Doctorado en Biología Marina, Facultad de Ciencias, Universidad Austral de Chile, Campus Isla Teja s/n, Valdivia, Chile
[2] Instituto de Ciencias Marinas y Limnológicas, Facultad de Ciencias, Universidad Austral de Chile, Campus Isla Teja s/n, Valdivia, Chile
[3] Centro FONDAP de Investigación en Dinámica de Ecosistemas Marinos de Altas Latitudes (IDEAL), Valdivia, Chile
[*] These authors contributed equally to this work.

Corresponding author
Sara M. Rodríguez,
saramrodriz@gmail.com

## ABSTRACT

**Background**. Parasites are essential components of natural communities, but the factors that generate skewed distributions of parasite occurrences and abundances across host populations are not well understood.

**Methods**. Here, we analyse at a seascape scale the spatiotemporal relationships of parasite exposure and host body-size with the proportion of infected hosts (i.e., prevalence) and aggregation of parasite burden across ca. 150 km of the coast and over 22 months. We predicted that the effects of parasite exposure on prevalence and aggregation are dependent on host body-sizes. We used an indirect host-parasite interaction in which migratory seagulls, sandy-shore molecrabs, and an acanthocephalan worm constitute the definitive hosts, intermediate hosts, and endoparasite, respectively. In such complex systems, increments in the abundance of definitive hosts imply increments in intermediate hosts' exposure to the parasite's dispersive stages.

**Results**. Linear mixed-effects models showed a significant, albeit highly variable, positive relationship between seagull density and prevalence. This relationship was stronger for small (cephalothorax length >15 mm) than large molecrabs (<15 mm). Independently of seagull density, large molecrabs carried significantly more parasites than small molecrabs. The analysis of the variance-to-mean ratio of per capita parasite burden showed no relationship between seagull density and mean parasite aggregation across host populations. However, the amount of unexplained variability in aggregation was strikingly higher in larger than smaller intermediate hosts. This unexplained variability was driven by a decrease in the mean-variance scaling in heavily infected large molecrabs.

**Conclusions**. These results show complex interdependencies between extrinsic and intrinsic population attributes on the structure of host-parasite interactions. We suggest that parasite accumulation—a characteristic of indirect host-parasite interactions—and subsequent increasing mortality rates over ontogeny underpin size-dependent host-parasite dynamics.

Subjects Animal Behavior, Biodiversity, Parasitology
Keywords Acanthocephalans, Laridae, Parasite aggregation, Temporal variation, Molecrabs

## INTRODUCTION

Parasites commonly show aggregated distributions across host populations. The degree of parasite aggregation—i.e., few hosts concentrate most of the parasites—can have consequences for the stability of parasite-host interactions (*Wilson et al., 2001*). For example, seminal theoretical work predicts that parasite aggregation stabilises host-parasite dynamics (*Anderson & May, 1978*; *May & Anderson, 1978*). However, extreme aggregation in host populations with low prevalence, such that a very small proportion of hosts carry a high parasite burden, might reduce the ability of the parasite to regulate the host population (*Anderson & May, 1978*). Variations in "extrinsic" factors such as exposure to infective stages of parasites (hereafter referred to as "parasite exposure") can generate parasite aggregation across hosts (*McCoy et al., 2016*). However, "intrinsic" host-population factors, such as age and body-size structure, can significantly influence the links between variation in parasite exposure and aggregation (*Anderson & May, 1978*). Determining the links of these extrinsic and intrinsic factors with patterns of prevalence and aggregation across host populations will improve our understanding of complex host-parasite dynamics (*Shaw, Grenfell & Dobson, 1998*; *Wilson et al., 2001*; *Morand & Krasnov, 2008*).

Variations in parasite exposure can affect infection rates and thus parasite aggregation across host populations. Assuming a certain, basal level of parasite aggregation (*Wilson et al., 2001*), an increase in parasite exposure will lead to an increase in the proportion of infected hosts (i.e., prevalence) and the parasite burden of the already-infected individual; this is because infection increases host susceptibility to subsequent infections when there is no further increase in host's immunity (*Wilson et al., 2001*). For example, the Pacific chorus frog *Pseudacris regilla* (Baird & Girard, 1852) shows increased susceptibility to the trematode *Ribeiroa ondatrae* Looss, 1907 after being infected, because the previous infection impairs hosts' immunological system (*Johnson & Hoverman, 2014*). Therefore, it can be hypothesised that increased parasite exposure will increase the infection rate of already-infected hosts, increasing parasite burden and therefore the aggregation across the population.

The effect of parasite exposure on aggregation can depend on host size structure. Assuming that body size is correlated with host age (*Ebert, 1999*), it can be suggested that small hosts have been exposed for a shorter time to parasite infection than larger conspecifics (*Muñoz & George-Nascimento, 2008*)—this in turn should result in smaller parasite burdens in the former than the latter (*Grutter & Poulin, 1998*; *Poulin, 2013*). For example, subpopulations of fish composed by larger individual have a higher prevalence and per capita parasite burden than subpopulation composed by smaller individuals (*Lo, Morand & Galzin, 1998*; *Poulin, 2000*; *Muñoz, Valdebenito & George-Nascimento, 2002*). Since infection is positively related with host susceptibility (e.g., *Johnson & Hoverman, 2014*), then it can be suggested that larger individuals will have larger infections rates than smaller individuals (see *Muñoz, Valdebenito & George-Nascimento, 2002*; *Muñoz & George-Nascimento, 2008* for an example in fish hosts). In complex parasite-host interactions, intermediate hosts are usually unable to expel the parasites and thus they accumulate

parasites over the lifetime (*Muñoz & George-Nascimento, 2008*; *Campião et al., 2015*). In addition, the probability of mortality can be larger for heavily infected hosts, leading to a decrease in prevalence and aggregation over ontogeny (*Rousset, 1996*; *Duerr et al., 2003*). We can expect that in such complex systems, therefore parasite exposure should have differential effects on the parasitosis of small- and large-sized intermediate hosts.

Migratory seagulls and their parasites constitute a model system to investigate how parasite exposure and intermediate host size-structure relate with prevalence and aggregation. Migratory Charadriiformes are usually involved in indirect parasite-host interactions, in which the birds are the definitive host and there is at least one intermediate host, usually invertebrates that accumulate parasites over their ontogeny (e.g., *Rodríguez, D'Elía & Valdivia, 2016*). In these interactions, parasite infective stages are trophically transmitted between definitive and intermediate hosts, so that variations in the abundance of the former imply significant variations in parasite exposure of the latter (*Latham & Poulin, 2003*; *Smith, 2007*). For example, local increments in the abundance of migratory birds have been linked to increases in the prevalence of infectious diseases in intermediate hosts (*Altizer, Bartel & Han, 2011*; *McCoy et al., 2016*). In addition, increases in seagull abundances can be followed by concomitant increments in per capita parasite loads in the intermediate host populations (*Latham & Poulin, 2003*; *Smith, 2007*). Accordingly, it can be expected that increments in the abundance of migratory seagulls should result in an increase in parasite exposure. If increasing exposure positively affects the mean infection rate, then the concomitant increase in the spatial variance in parasite infection can increase parasite prevalence and aggregation across intermediate host populations. Migratory Charadriiform species are under severe threats worldwide, owing to human-induced habitat destruction, hunting, and climate change-related impacts on migratory timing, among other factors (*Munro, 2017*). Understanding how migratory Charadriiformes take part in complex host-parasite systems can have therefore important implications for both, fundamental and applied ecology.

In this study we test the hypothesis that, if the variability in parasite exposure and host body size generate intra-population variability in host infection rates, both factors interdependently influence prevalence and aggregation across host populations. We predicted that increases in migratory seagull densities should be linked to increases in parasite prevalence and aggregation in intermediate large-, but not small-sized, intermediate hosts. An indirect host-parasite interaction, composed by migratory seagulls (definitive hosts), molecrabs (intermediate hosts), and an acanthocephalan worm (parasite), was analysed. In this complex system, infected seagulls release the infective stage of *Profilicollis altmani* Meyer, 1931 in their faeces. These faeces are then ingested by the molecrab *Emerita analoga* Stimpson, 1857, and the cycle is completed after predation on molecrabs by the seagulls *Larus dominicanus* (Lichtenstein, 1823), *Chroicocephalus maculipennis* Lichtenstein, 1823, *Leucopheus modestus* (Tschudi, 1843), and *Leucopheus pipixcan* (Wagler, 1831; *Rodríguez, D'Elía & Valdivia, 2016*).

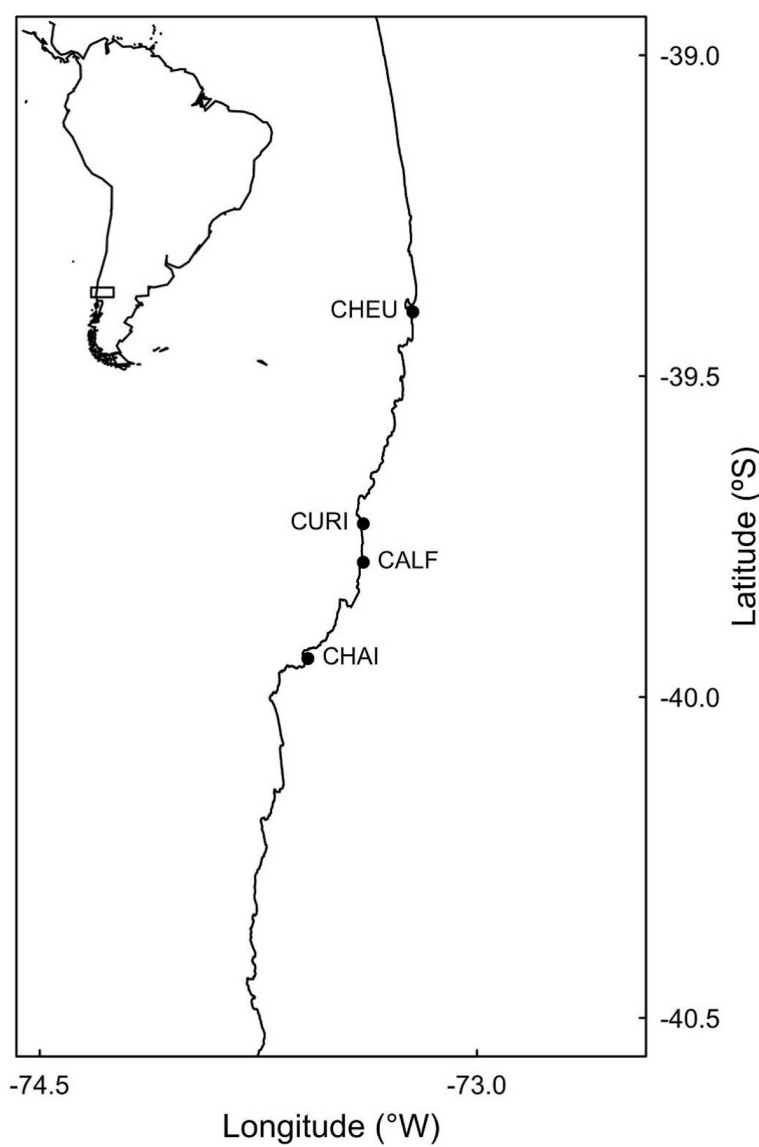

**Figure 1 Study sites.** Site codes are CHEU, Cheuque; CURI, Curiñanco; CALF, Calfuco; and CHAI, Chaihuín.

## MATERIALS AND METHODS

This study complies with the law No 19.473 of Servicio Agrícola y Ganadero of Chile, which permits observational studies on any open access coastal area. We sampled four sandy beaches located in the central-southern Chilean shore. The sites were Cheuque, Curiñanco, Calfuco, and Chaihuín, and spanned ca. 150 km of the southeast Pacific shore (Fig. 1). Three of the four seagulls analysed, namely *C. maculipennis, L. modestus* and *L. pipixcan*, use these shores as feeding habitats during the austral summer. *Larus dominicanus* can be observed in the area throughout the year. These populations display large proportions of individuals infected with *P. altmani*. On average, the individuals of the four species show comparatively large numbers of *P. altmani* parasites in the intestine (see Table 1 in

*Rodríguez, D'Elía & Valdivia, 2016*). Previous records studies have described *Profilicollis* larvae in intestines of other vertebrate hosts, such as sea otters (*Hennessy & Morejohn, 1977*; *Margolis et al., 1997*; *Mayer, Dailey & Miller, 2003*). However, no adult parasites have been recorded in these hosts (*Mayer, Dailey & Miller, 2003*), so the probability that sea otters encompass an alternative vector of infection is still low.

Densities of seagulls and *E. analoga* were quantified trimonthly between January 2014 and October 2015 (i.e., 22 months). This sampling period allowed us to analyse almost two full seasonal cycles of seagull migrations. In each bird census, each sandy beach was sampled in full during low tide—samplers walked parallel on effluent line to avoid disturbing the seagulls. A prismatic lens, with a spotting scope between 20 and 60×, was used to quantify the number of individuals of the four seagull species (*Navedo et al., 2015*). The length of each beach was determined using a GPS and the Google Earth freeware software. The abundance of seagulls was expressed as the kilometric abundance index (ind km$^{-1}$). Additionally, we quantified the number of people, dogs, and motorised vehicles (i.e., "visitors") in each site and sample time as a potential factor influencing bird abundance. Preliminary regression analyses revealed no significant relationship between the total number of visitors and bird abundance across sites and over time ($R^2 = 0.002$; $P > 0.05$. Fig. S1).

To survey the density of molecrabs, in each site and sampling time we deployed four transects randomly and perpendicularly to the shoreline. In each transect, we placed four sampling stations located ca. 2 m apart from each other from the effluent line to the swash line. Plastic corers (0.03 m$^{-2}$) were buried to a depth of 20 cm (equating a volume of 0.006 m$^{-3}$) and the sand was sieved through a 1-mm mesh sieve. Cores were combined into transects in order to integrate all molecrab sizes in each sampling unit. Molecrabs were transferred to the laboratory for parasite identification and sorting under binocular microscope. In the laboratory, cephalothorax length (mm) of each individual was recorded and then they were dissected in order to extract the *P. altmani* cystacanths (i.e., larval stage) from the haemocoele. The host population was composed by two subgroups of body sizes, one ranging from 0.65 to 15 mm and a second group ranging from 15 to 32 mm (Fig. 2A). For brevity, the former and the latter will be referred to as "small" and "large" hosts, respectively.

For each size group, site, and sampling time, prevalence and parasite burden were separately estimated as the proportion of infected hosts and the per capita number of parasites, respectively (*Bush et al., 1997*). Parasite aggregation was estimated in two ways, as the variance-to-mean ratio and the scalar exponent of Taylor's power law (*Taylor, 1961*). The variance-to-mean ratio was estimated as the quotient between the variance ($s^2$) and mean ($m$) of the per capita parasite burden. This ratio varies from zero (indicating that parasites are uniformly distributed among hosts) through one (indicating a random distribution of parasites), to a maximum bounded by the number of parasites in the sample (indicating an aggregated distribution of parasites; reviewed in *Wilson et al., 2001*). The scalar exponent $b$ of Taylor's power law was estimated from $s^2 = am^b$ (*Taylor, 1961*); where $a$ is the variance when $m = 1$ and $b$ is the mean–variance scalar exponent. The $b$ parameter was estimated as the slope of the natural log–log regression between $s^2$ and $m$ (Figs. S2 and S3).
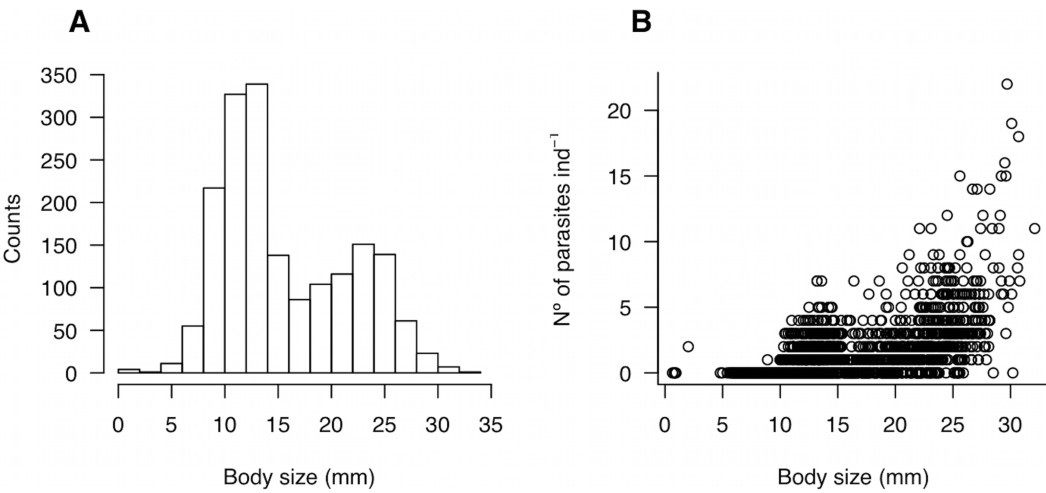

**Figure 2** (A) Size-structure of the molecrab *Emerita analoga* and (B) relationship between body size (mm) and parasite burden (number of parasites ind$^{-1}$).

The scalar exponent varies from zero (indicating a uniform distribution of parasites) through one (for a random, Poisson, distribution of parasite burden), to infinity (highly aggregated distribution of parasite counts). Since the intercept ($a$) of this regression can vary significantly from zero with small $m$ values, we took into account the former by estimating $b$ at a standard mean of $\log_{10} m = 1$. This estimator is called $I_{10}$ and is useful to disentangle the patterns of aggregation from those of mean parasite load (*Lester, 2012*; *Lester & Mc Vinish, 2016*). We estimated bias-corrected confidence intervals for each site after 1,000 bootstrapped estimations of $I_{10}$. To further assess the relationship between mean parasite load and $b$, the fits were repeated for two subpopulations of $m = [0.9, 1.59]$ and $[1.6, 6.1]$. Prevalence and aggregation were estimated as attributes of each site and sampling time, over transects within each site. These estimations were conducted separately for each size group. The $b$ and $I_{10}$ indexes were estimated as attributes of each site.

## Statistical analyses

The prediction of size-dependent relationships of parasite prevalence and aggregation with seagull density was tested by means of linear mixed-effects models (LMM). In these models, molecrab body size group and bird density were fixed and crossed variables, and parasite prevalence and aggregation (variance-to-mean ratio) were the dependent variables. Site and sampling time (nested in site) were included in the models as random factors. For each LMM model, we estimated pseudo-$R^2$ according to *Nakagawa & Schielzeth (2013)* and *Johnson (2014)*. Two types of pseudo-$R^2$ were calculated for each model: the marginal $R^2$, which represents the variance explained by the fixed factors, and the conditional $R^2$, which represents the variance explained by both, fixed and random factors. The models were fit through Restricted Maximum Likelihood (REML). The degrees of freedom of each fixed effect term (bird density, molecrab size class, and interaction) were estimated through the Satterthwaite approximation, and then hypothesis test was done on the basis of conditional $t$-tests (*Pinheiro & Bates, 2004*). The treatment contrast was used to

**Table 1** Number of small and large crabs of *E. analoga*, number of parasites in small and large molecrabs (SM and LM) and number of seagulls for each sample seasons during 2014 and 2015.

| | Su-2014 | Au-2014 | Wi-2014 | Sp-2014 | Su-2015 | Au-2015 | Wi-2015 | Sp-2015 |
|---|---|---|---|---|---|---|---|---|
| Small molecrabs | 212 | 121 | 133 | 72 | 132 | 217 | 81 | 73 |
| Parasites SM | 117 | 102 | 83 | 83 | 89 | 157 | 49 | 56 |
| Large molecrabs | 131 | 100 | 76 | 49 | 74 | 168 | 72 | 69 |
| Parasites LM | 245 | 244 | 141 | 165 | 217 | 513 | 211 | 257 |
| Seagulls | 867 | 1,564 | 394 | 485 | 903 | 472 | 110 | 71 |

**Notes.**
Su, Summer; Au, Autumm; Wi, Winter; Sp, Spring.

estimate effect coefficients. Standardised residual-vs.-fitted values plots were used to check homogeneity of variances in LMMs. Variance-to-mean ratio data were $\log_{10}$-transformed to homogenise the variances between both size groups (see Results). The visual analysis of residuals suggested that a Gaussian model was the most appropriate for both, prevalence and $\log_{10}$ variance-to-mean ratio. Autocorrelation functions were used to check the temporal autocorrelation of residuals.

In order to determine if the potential relationships of bird density with parasite prevalence and aggregation were lagged over time (for example, the increase in bird density would result in an increase of parasitosis three months later), we computed cross-correlation functions (CCF) between bird density (x) and prevalence (y) and between the former and the variance-to-mean ratio (y)—we additionally cross-correlated bird and molecrab abundances. All statistical analyses were conducted in R 3.3.0 (*R Development Core Team, 2012*). Linear mixed-effects models, pseudo-$R^2$, and CCF were computed with the *lme4, MuMIn,* and *stats* libraries, respectively (*Bates et al., 2015*; *Bartón, 2016*; *R Development Core Team, 2012*).

## RESULTS

A total of 1,780 individuals of *E. analoga* were captured throughout the study. Molecrab cephalothorax length varied from 0.65 to 32.1 mm, with two modes located at 11.6 and 23.7 mm respectively (Fig. 2A). Parasite burden—i.e., the per capita number of parasites—ranged between zero and 22 ind. per molecrab (Fig. 2B). In addition, parasite burden tended to increase with increasing molecrab' cephalothorax length (Fig. 2B; Table 1).

Seagull abundance (i.e., kilometric abundance index) varied between 11 ind km$^{-1}$ in winter, to 906 ind km$^{-1}$ in summer and 922 ind km$^{-1}$ in autumn (Fig. 3A). Molecrab density ranged between 14 ind. 0.006 m$^{-3}$ and 167 ind. 0.006 m$^{-3}$ in summer and spring, respectively (Fig. 3B). Bird and small-sized molecrab abundances showed instantaneous (i.e., no lagged) correlations in Cheuque (Fig. S4). In addition, bird abundance was correlated with both, small and large molecrabs at three-month lag in Calfuco (Fig. S4).

Site-level prevalence values were 44 ± 18% (mean ± standard deviation) and 79 ± 15% for small and large molecrabs, respectively (Figs. 4A and 4B). For both groups, prevalence was highly variable across sites and over time, with maximum values occurring in summer 2015 (small molecrabs, Fig. 4A) and winter 2014 (large molecrabs, Fig. 4B). Despite these

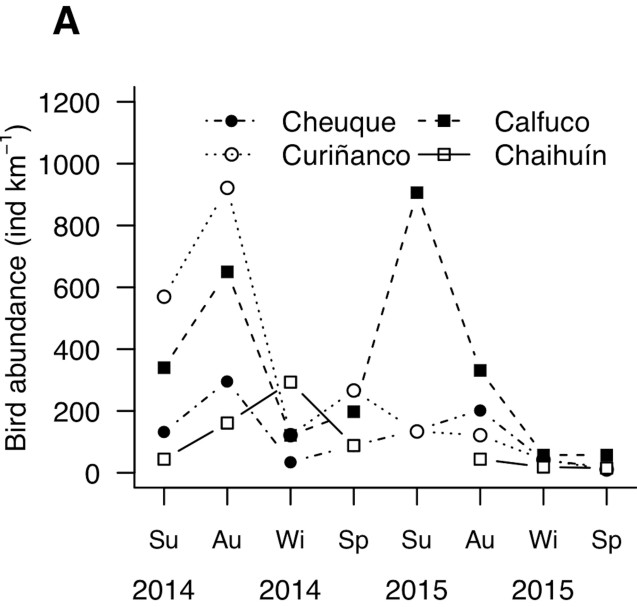

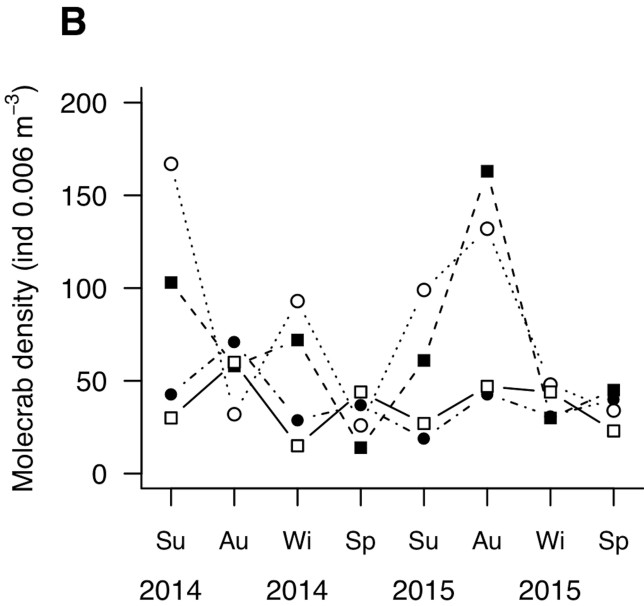

**Figure 3** **Mesoscale spatiotemporal patterns of (A) bird and (B) molecrab abundances during 2014 and 2015.** Su, summer; Au, autumn; Wi, winter; Sp, spring. Symbols indicate the study sites.

variable patterns, the analysis of LMM detected a significant relationship between bird abundance and prevalence (Fig. 4C, Table 2). In addition, large molecrabs showed ca. 40% higher prevalence than the small molecrabs (Fig. 4C, Table 2). The slope between bird density and prevalence tended to be steeper for small than large molecrabs (Fig. 4C), albeit this difference was not statistically significant at alpha = 0.05 (Table 2). The fixed factors

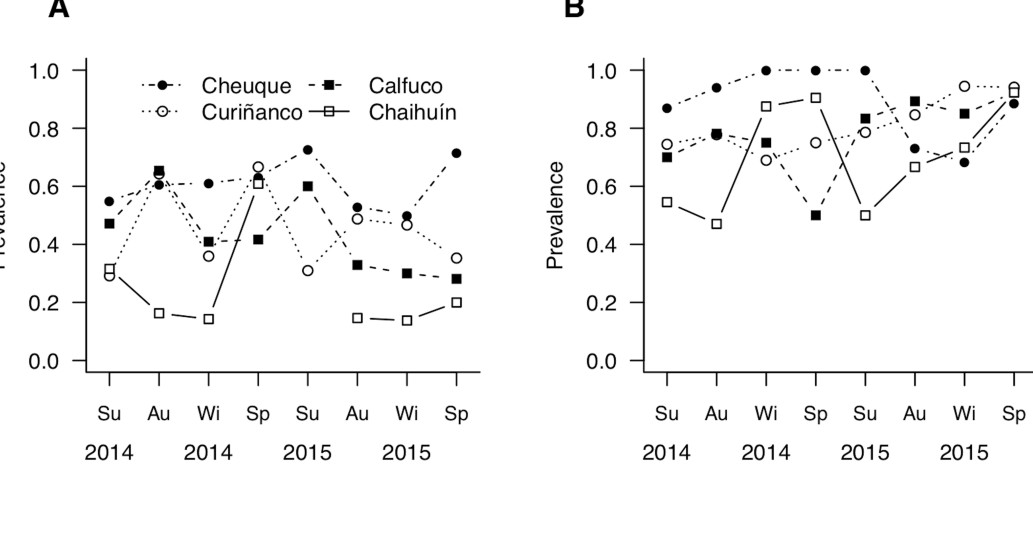

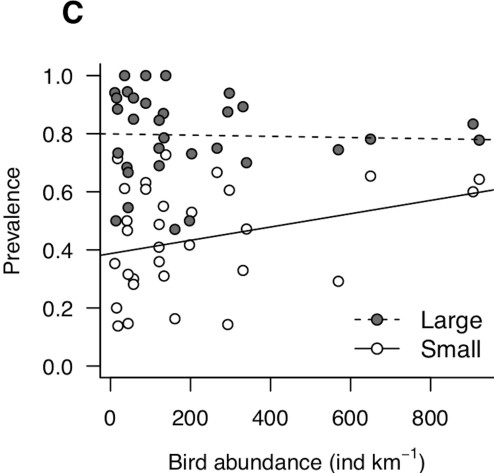

**Figure 4** **Patterns of parasite prevalence in population fractions composed by (A) small and (B) large hosts during 2014 and 2015.** Su, summer; Au, autumn; Wi, winter; Sp, spring. (C) Relationship between bird abundance and prevalence; the statistical fits between both variables is shown adjusted for host body size (see coefficients in Table 2).

in the model accounted for ca. 54% of the variability in prevalence, while the entire model accounted for ca. 76% (Table 2).

Parasite aggregation tended to be greater in large than small hosts across sites and over time (Figs. 5A and 5B). In addition, the uncertainty was also greater in large than small hosts, evidenced by the broader ranges in aggregation observed in the former (Figs. 5A and 5B). This wide and size-dependent variation led to a statistically non-significant relationship of aggregation with bird density and molecrab body size (Table 2). The analysis of aggregation at a fixed mean parasite load of $m = 1(I_{10})$ suggested, however, that the apparent greater aggregation and stochasticity in large molecrabs was due to heterogeneities in mean parasite load, as $I_{10}$ scores remained similar across sites (Figs. 5C and 5D). The small-sized molecrab subpopulations showed $I_{10}$ scores between 1 and 2

**Table 2  Results of LMM for the relationships of parasitosis (prevalence and variance-to-mean ratio of parasite burden) with bird abundance and molecrab body size.** Estimate effects, their standard errors (S.E.), degrees of freedom (D.F.), and $t$ statistics are provided for each fixed effect term. Degrees of freedom were estimated through Satterthwaite approximation. $R^2m$ and $R^2c$ are the marginal and critical pseudo-$R^2$, which represent the variability accounted for fixed and the entire model, respectively.

|  | Term | Estimate | S.E. | D.F. | $t$-value | $P$-value | $R^2m$ | $R^2c$ |
|---|---|---|---|---|---|---|---|---|
| Prevalence | Intercept | 0.3871 | 0.0624 | 4.54 | 6.203 | 0.002 | 0.538 | 0.757 |
|  | Bird density | 0.0002 | 0.0001 | 53.36 | 2.150 | 0.036 |  |  |
|  | Effect of large molecrabs | 0.4120 | 0.0416 | 29.39 | 9.913 | <0.001 |  |  |
|  | Interaction | −0.0002 | 0.0001 | 28.93 | −1.896 | 0.068 |  |  |
| Variance-to-mean ratio ($\log_{10}$) | Intercept | 0.1843 | 0.0538 | 18.32 | 3.4287 | 0.0029 | 0.138 | 0.154 |
|  | Bird density | −0.0001 | 0.0002 | 58.32 | −0.7427 | 0.4607 |  |  |
|  | Effect of large molecrabs | 0.0726 | 0.0719 | 56.04 | 1.0096 | 0.3170 |  |  |
|  | Interaction | 0.0004 | 0.0002 | 55.98 | 1.5970 | 0.1159 |  |  |

across sites (Fig. 5C). The large molecrabs showed slightly larger $I_{10}$ scores in Cheuque and Chaihuín (Fig. 5D). These results point to a general Poisson distribution of parasites in the study region.

In addition, $b$ depended on mean parasite load, as large-sized molecrabs showed a statistically significant $b = 1.1$ (0.4) when $m < 1.6$, but non-significant $b = 0.9$ (0.7) when $m \geq 1.6$ (standard errors bracketed). Prevalence and aggregation (variance-to-mean ratio) did not show lagged correlations with bird density or molecrab abundance across host sizes—the lack of correlation of molecrab abundance with prevalence indicates that the former values are comparable between sites, times, and size groups (Figs. S5–S8).

## DISCUSSION

This study showed complex relationships of parasite exposure, represented by definitive host abundances, with prevalence and aggregation across intermediate host population. Acanthocephalan prevalence across molecrab populations was significantly related with bird abundance, suggesting a link between parasitosis and parasite exposure. However, this relationship was relatively inconsistent, as random effects of temporal surveys and sites accounted for ca. 20% of the variation in prevalence. In addition, we observed that prevalence in large (>15 mm) molecrabs was significantly larger than in small (<15 mm) molecrabs. Over seasonal surveys and across sites, parasite aggregation remained relatively constant for small, but highly variable and unpredictable for large molecrabs. However, these latter patterns were largely driven by variations in mean parasite load, rather than actual aggregation. Moreover, further analyses of Taylor's power law showed that greater mean parasite loads were associated with slower increases in the load variance with the mean of large molecrabs. Below, we discuss how parasite accumulation over the ontogeny and differential mortality rates across host populations can influence the links among parasite exposure, intermediate-host size structure, and parasitosis.

We observed a significant positive relationship between prevalence and body size of the intermediate hosts. Similarly, the analyses indicate heavier parasite load in large individuals. Parasite accumulation over time could explain these trends. The analysed host-parasite

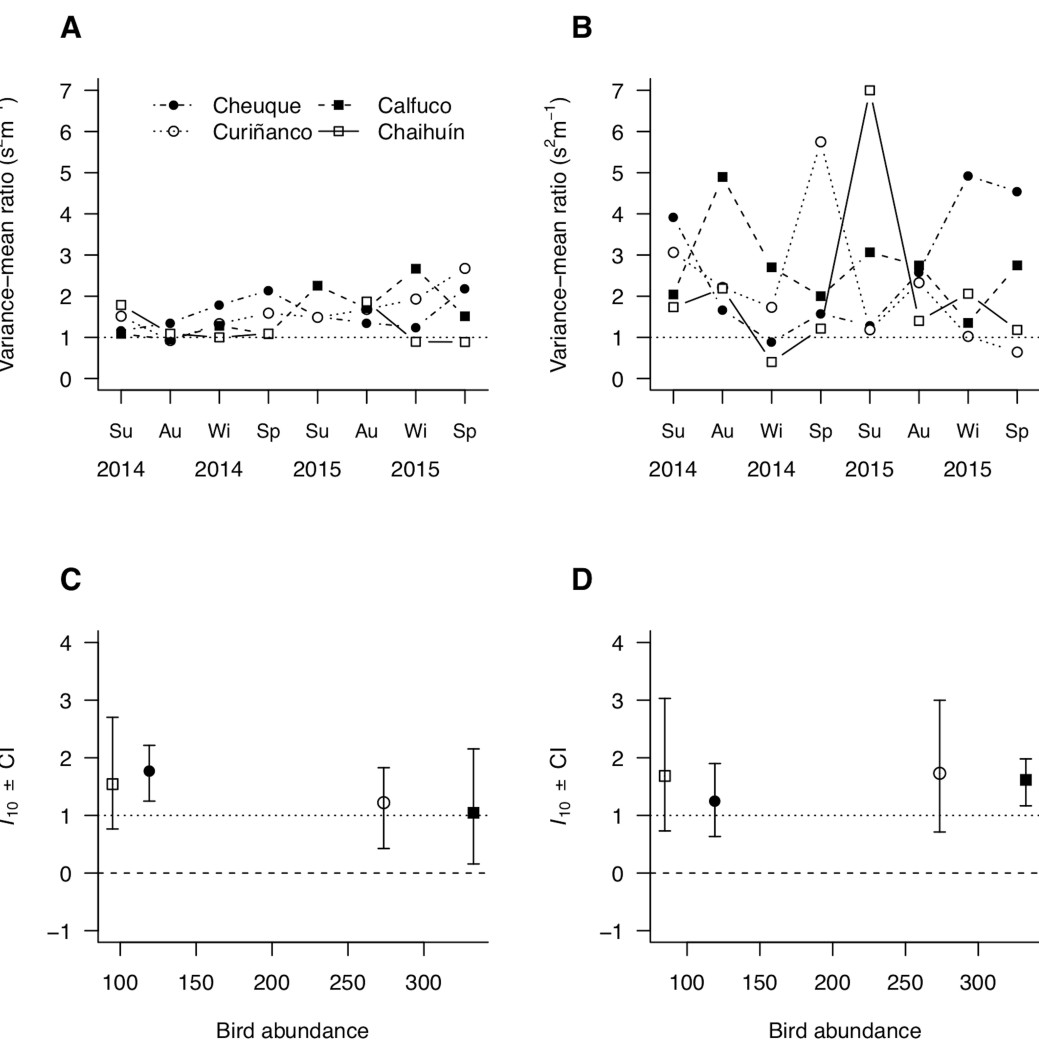

**Figure 5** Spatial patterns of parasite aggregation estimated as (A, B) the variance-to-mean ratio and (C, D) the scalar coefficient of Taylor's power law for small (A, C) and large (B, D) molecrabs.

interaction is characterised by the inability of hosts to expel the parasites once they are allocated in the haemocoele (*Latham & Poulin, 2003*). Accordingly, larger molecrabs, which have been exposed for longer time to successive infections, accumulate parasites over the ontogeny. Similar processes have been described for other endoparasites, like *Profilicollis antarcticus* Zdzitowiecki, 1985 and *P. novaezelandensis* Brockerhoff & Smales, 2002, which accumulate over the ontogeny of Ocipodidae and Grapsidae crabs (*Latham & Poulin, 2003*). Similarity, the prevalence, intensity, and taxonomic richness of a number of endoparasite groups can increase during the ontogeny of fish hosts (*Muñoz, Valdebenito & George-Nascimento, 2002*). Moreover, body size of anuran hosts can be a strong predictor of the richness of helminthic fauna—this pattern is consistent across hosts' geographic ranges and phylogenetic correlations, in addition to sampling effort (*Campião et al., 2015*). Accordingly, parasite accumulation seems to be a widespread characteristic of indirect host-parasite interactions.

In our study, size-dependent mortality might explain the significant decrease in the scalar coefficient of Taylor power law with increasing mean parasite load in the large-sized molecrab subpopulation. *Latham & Poulin (2002a)* and *Latham & Poulin (2002b)* indicate that high parasite burden of two acanthocephalan species increase the mortality rate in largest crab size-classes. Acanthocephalan species are non-lethal parasites, but they can alter host's behaviour and increase their susceptibility to predation (*Latham & Poulin, 2002a*; *Latham & Poulin, 2002b*; *Beisel & Medoc, 2010*). For example, molecrab' burrowing velocity is negatively correlated with parasite burden (*Jerez & George-Nascimento, 2010*). In addition, egg mortality during dispersion could also have led to the non-significant relationship between bird density (i.e., exposure to parasite infection) and the comparatively low levels of aggregation observed in this study. Abiotic environmental variables related with emersion, such as desiccation and heat stress can affect negatively parasite' egg viability (*Herrmann & Poulin, 2011*; *Studer & Poulin, 2012*; *Studer, Lamare & Poulin, 2012*). Molecrab life history would offer a third explanation for the uncertainty in parasite load of large-sized host. We observed that molecrab recruitment strongly varied across geographic locations, with site-dependent peaks occurring in summer, autumn, and spring (but see *Contreras, Defeo & Jaramillo, 1999*). An increase in non-infected recruits can have negative (positive) effects on prevalence (aggregation) across the host population (e.g., *Zambrano & George-Nascimento, 2010*), which might decouple the relationship between exposure and the parasitosis in the intermediate host. Further field-based comparative experiments would be useful to tease apart the effects of exposure, egg mortality, and recruitment timing on parasite aggregation.

To what degree these results could be generalised to other host-parasite systems? The life cycles of almost all acanthocephalan and digenetic species, as well as some nematodes and cestodes, include at least one intermediate host that accumulates parasites over time (*Thomas, Guégan & Renaud, 2009*). In addition, size-dependent responses of prevalence to parasite exposure have been demonstrated in other complex host-parasite interactions involving migratory birds and crustaceans (*Latham & Poulin, 2003*; *Smith, 2007*; *Zambrano & George-Nascimento, 2010*). In the same line, parasite burden increases with body size of the toadfish *Aphos porosus* Valenciennes, 1837, an intermediate host of cestodes, nematodes, trematodes, and acantocephalans (*Cortés & Muñoz, 2008*). These examples suggest that accumulation of endoparasites in intermediate hosts may be a common phenomenon. Contrarily, ectoparasites, which can be removed by hosts, can show different patterns of prevalence and aggregation. In such systems, the smaller or younger hosts accumulate greater parasite burdens than older hosts, as a likely consequence of higher susceptibility to infections of the former than the latter (*McCoy et al., 2016*). Therefore, significant relationships between parasite exposure and prevalence, in addition with size-dependent patterns of stochasticity in parasite load could be observed in other complex host-parasite systems, at least those involving endoparasites that accumulate over host' ontogeny.

In summary, our results showed that parasitosis could be related with exogenous (exposure) and endogenous (host body size) factors in a complex fashion. Parasite exposure was positively related with prevalence in the intermediate host population. Prevalence and parasite load were dependent of intermediate host body size, which may be explained

by parasitosis-related increasing mortality rates over host' ontogeny. Also, these results would be generalised to other complex parasite with an intermediate host that accumulates parasite over time. These results suggest that intrinsic host-population characteristic can have stronger effects on the dynamic of host-parasite interactions than extrinsic factors. Overall, our study contributes to the understanding of the processes that underpin the heterogeneity in parasite burden across host populations.

## ACKNOWLEDGEMENTS

We thank all undergraduate interns (Biología Marina; UACH) who provided invaluable help during the fieldwork. Daniela López contributed with helpful suggestions and discussions that improved previous versions of this manuscript. This paper belongs to SMR PhD thesis dissertation at Universidad Austral de Chile.

### Funding

This work was supported by Fondo Nacional de Desarrollo Científico y Tecnológico (No. 1141037), by Beca Doctorado Nacional and Gastos Operacionales CONICYT (No. 21120304), by FONDAP-IDEAL (No. 15150003) and by the Dirección de Investigación y Desarrollo, Universidad Austral de Chile (DID). The funders had no role in study design, data collection and analysis, decision to publish, or preparation of the manuscript.

### Grant Disclosures

The following grant information was disclosed by the authors:
Fondo Nacional de Desarrollo Científico y Tecnológico: 1141037.
Beca Doctorado Nacional and Gastos Operacionales CONICYT: 21120304.
FONDAP-IDEAL: 15150003.
Dirección de Investigación y Desarrollo, Universidad Austral de Chile (DID).

### Competing Interests

The authors declare there are no competing interests.

### Author Contributions

- Sara M. Rodríguez conceived and designed the experiments, performed the experiments, contributed reagents/materials/analysis tools, wrote the paper, prepared figures and/or tables.
- Nelson Valdivia conceived and designed the experiments, analyzed the data, contributed reagents/materials/analysis tools, wrote the paper, prepared figures and/or tables, reviewed drafts of the paper.

### Animal Ethics

The following information was supplied relating to ethical approvals (i.e., approving body and any reference numbers):

Our study complies to Law No 19.473 of Servicio Agrícola y Ganadero of Chile, which permits conducting purely observational research without any additional approval document.

## Data Availability

The raw data has been uploaded as a Supplemental File.

## Supplemental Information

Supplemental information for this article can be found online at http://dx.doi.org/10.7717/peerj.3675#supplemental-information.

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
