# Peer review of "Mesoscale spatiotemporal variability in a complex host-parasite system influenced by intermediate host body size"

_PeerJ, doi:10.7717/peerj.3675_

## Round 0.1 · original submission · Major Revisions

Please revise your manuscript, paying close attention to the reviewers' concerns. Be advised that your revised paper will probably be sent out for a second review.

·

Basic reporting

Generally good. The supplementary material could use a key to headings.

Experimental design

Better measure(s) of aggregation required.

Validity of the findings

Validity will benefit from additional analysis.

Additional comments

Interesting paper with great data set. The headings on the supplementary sheet could be made clearer, or perhaps a separate key provided to explain the meanings of the headings. Does the column headed ‘N.P. bullocki’ contain the numbers of Profilicollis altmani, for example?

The authors sought to test whether gull abundance was linked to the prevalence and aggregation of an acanthocephalan parasite in large but not small crabs, the crabs being the intermediate hosts for the parasite and the gulls the definitive hosts. They concluded that there was no strong link but that there was a large amount of unexplained variability in aggregation especially in large intermediate hosts.

I think some reanalysis of their data might be useful as there may be better methods to measure the degree of aggregation.

Specific points:
1. Are gulls the only primary hosts in all four areas? I note that even sea otters carry the infection.

2. Lines 156+ Two methods were used to measure aggregation, the variance/mean ratio (otherwise known as the Index of Dispersion) and the slope of Taylor’s Law. In most parasites, the Index of Dispersion is dependent on the size of the mean. A high index is typical of a heavily infected sample. It does not indicate that the parasite is more aggregated in that sample. A better method to measure aggregation might be something like Poulin’s D (Poulin, 1993).

The second method the authors used, the slope in Taylor’s Law, is better than the variance/mean ratio as it is less dependent on the mean but is not very satisfactory because it does not take into account the intercept, which is typically very significant with small means such as here. An alternate approach (the best I think but I could be biased) is to use Taylor’s Law to estimate the variance at a standard mean and use this to calculate the Index of Dispersion (e.g. Lester, 2012; Lester & MacVinnish, 2016). If I am interpreting the supplementary data correctly the overall I10 appears to be around 3, i.e. not very aggregated. The lack of aggregation considering the eggs are deposited in clumps may be worth discussing.

Reanalysis of the data with an improved measure of aggregation my reveal differences, or it may show that all along the coast at points 500 km apart the parasite shows the same pattern of aggregation.

3. Lines 253. The authors discuss possible host mortality associated with the parasite. Evidence for this could be looked for in the data, using one of several methods. For example is there any evidence the rate of increase of the variance slows at high means suggesting that heavily infected hosts are missing from the samples? Could negative binomials, truncated at various points, be fitted to the data? A literature search on methods for measuring parasite associated mortality and the application of one or two may help.

4. A summary table giving: numbers of large crabs, numbers of small crabs, season, and mean no. parasites for each of the sites sampled would help readers. Perhaps bird abundance could be included.

The paper is well worth publishing even if it shows that there was no change in aggregation with size, or with locality. The weak link with gull abundance is also worth recording. The supplementary tables are especially valuable as they contain excellent baseline data against which future studies can be compared.

Reviewer 2 ·

Basic reporting

I found the manuscript to be very clearly written. There were a few places where I thought some additional literature could be cited that showed other sides of the arguments that the authors were making (discussed below in Minor Comment), but generally I found the writing and discussion very good. The raw data was supplied.

Experimental design

The experimental design was good and I was impressed with the authors consistent sampling over time. I did have a number of questions on the GLMM used by the authors as listed below.

1. It was not immediately clear in the paper how the authors aggregated over samples within a transect and over transects. I am guessing prevalence/aggregation was calculated over all sampling points and transects for a given time and site? It would make the analysis easier to understand if this was made explicit. Moreover, when aggregation/prevalence was calculated for different body sizes, did the authors then calculate prevalences/aggregations within small and large body size groups within transects and sites? One or two sentences would help make this much more clear.

2. I do not think the authors give enough information regarding their GLMM analysis. Given the author were dealing with prevalence and aggregation data I am assuming they did not specify that the random component of the GLMM followed a normal distribution. If not, what distributions were chosen for the random distribution (e.g. Binomial and Gamma)? If the authors did use a normal distribution, they should justify this choice (i.e. were transformations used to obtain normality). Moreover, if the authors used a non-normal random component they should specify what link function they used (e.g. logit, log, etc.). This description could be easily added to the supplementary material and would greatly clarify the analysis. If the authors could include the code they used as well that would be very helpful (thank you for including the data!).

3. This comment is particularly for the prevalence analysis. Did the authors try an analysis in which their dependent variable was whether a particular individual molecrab was infected or not infected (0 or 1) and their fixed effects were bird density, the measured mole crab size for that individual, and interactions and their random effects were site, time, and transect? The random component of the GLMM would be Binomial. This would avoid having to collapse the data and naturally account for "prevalence" values that were computed from different sample sizes. As is, it is not entirely clear to me how the authors current analysis accounts for the fact that some prevalences are calculated from different sample sizes. Perhaps a weighting terms was used. Again I think this needs to be made more clear. Including the analysis code would clarify many of these questions.

4. For the aggregation GLMM, the authors specify that the variance in aggregation was larger for large bodied molecrabs than small bodied molecrabs. Did the authors account for this variance structure in their model? They note that this wide size-dependent variation led to a statistically non-significant relationship of aggregation. nlme provides relatively easy ways to model this variance (e.g. using varFunc and the weights argument). This would be a more robust way to account for this variance and strengthen the author's conclusions.

5. For readers that are unfamiliar with Taylor's Power Law a plot of log mean vs log variance for the data given here would be very useful. This would also show how many data points were used to calculate this slope which would help answer my question above regarding how exactly the authors went from the data sheet included with this submission to the data "collapsed" data they used in their analyses.

6. Overall, I think all of the analyses given here could be presented more clearly.

Validity of the findings

The validity of the findings are hard to address given the lack of clarity in the statistical analyses (described above). I think the premise of the paper is very interesting, but without more detail on how the analyses were conducted and why they were conducted that way it is hard to for me to feel completely confident about the results. This is by no means saying they are "invalid", but with the information that is currently given their validity is hard to assess.

There was one particular finding that I was curious about. The authors show that "Large" molecrabs have more variability in aggregation than "small" crabs. The authors used the criteria that Large is > 15 and and Small is < 15. Looking at the authors data, the "Large" group has a larger variance(body size) / mean(body size) ratio than the small group. Do the authors think that the result that there is increased variability in aggregation in Large hosts is strictly due to how they split the data?

Additional comments

Line 28: Change to "in such complex systems"

Lien 29: Change to "parasite's"

Line 30: Change "lineal" to "linear"

Lien 48: Remove "relevant"

Line 53: In the canonical Anderson and May model, increasing aggregation (decreasing k) always increases the stability of the host-parasite equilibrium, but decreases the region in which the equilibrium can exist. I suggest changing the end of this line to "..and reduce the ability of the parasite to regulate the host population (Anderson and May 1978)."

Line 65 - 66: This statement is true when there is no concomitant immunity. The authors should make this distinction.

Line 73-87: I like this paragraph and agree with everything that is said in it and the citations that the authors use. However, I think the authors should briefly acknowledge work that discussed how age/body-size intensity profiles can show humped patterns (Wilson et al. 2001 mentions this as well as theory by Rousset et al. 1996, Ecology, and Duerr et al. 2003). The authors mention this later in the discussion, but I think with one or two added sentences the authors can make it clear to the reader that while a lot of the times we might expect increases mean intensity with age/body size, this is not always the case.
Line 81: Do you mean host susceptibility?

Line 88: Could you rephrase this as "Migratory Charadriiformes (e.g. seagulls)..."?

Line 102: Increasing exposure doesn't necessarily lead to increasing aggregation. For example, under a simple death-immigration model (where the "immigration" rate is the exposure rate) the parasite distribution at any time t is Poisson. Increasing immigration rate increases the variance, but the variance to mean ratio (i.e. aggregation) is still fixed at 1 (Anderson and Gordon, 1982, Parasitology; Rousset et al. 1996, Ecology). However, increasing variability in exposure leads to increased aggregation. I'd suggest the authors more carefully state this expectation.

Line 111: Similar to above, I think the authors can be more explicit as to why they predict increased aggregation with increased mean seagull density. Is it because increasing mean seagull density has a corresponding increased variance in seagull density (I am just guessing based on Taylor's Law) such that parasite exposure really is more variable under higher mean densities such that we could expect increased aggregation?

Line 135: If this prismatic lens is standard practice, a citation here would be useful.

Line 167: The negative binomial distribution can produce a max Taylor's power law slope of 2, the referencing the negative binomial in parentheses as an example of an infinite slope is misleading.

Line 171: Should be "Generalized linear"

Line 173: Should be "Dependent variables"

Line 176: How did you chose this error structure? Did an ACF of the residuals suggest that this was an appropriate model for the residuals? Was this a better model than not including an auto-regressive error structure? A bit more justification here would be helpful.

Line 186: Add in package citations for nlme and MuMIn.

Line 199: Do you mean 0.03 m^{-3}?

Line 203: Fig S2. Could the authors specify in the legend for Fig S2. that lag = 1 is equivalent to 3 months?

Line 214: Replace "stronger" with "greater" or "larger"

Line 217-219: How many points went into calculating this TPL coefficient? A figure showing the Taylor's Law from this data (log mean vs log var) plot would make it more clear where this b value was coming from.

Line 281-287: I like this discussion.

Table 1: I think that the sample size needs to be mentioned somewhere in this caption. Also, how were the p-values calculated? Depending on how you do it, the "traditional" p-values for GLMMs can be anti-conservative (http://bbolker.github.io/mixedmodels-misc/glmmFAQ.html#why-doesnt-lme4-display-denominator-degrees-of-freedomp-values-what-other-options-do-i-have). It would be good for the authors to state how they got these p-values and, if necessary recalculate them based on the recommendations in the above link. Also, since molecrab size is a factor, I think the coefficient given is the effect of large molecrabs on prevalence relative to small molecrabs (a sum to zero constraint). If so I think "Molecrab size" should be relabeled to "Effect of large molecrabs".

---

## Round 0.2 · accepted · Accept

Thank you for your careful revisions.

·

Basic reporting

Fine

Experimental design

Fine

Validity of the findings

OK

Additional comments

This is a stimulating manuscript, not least because it leads the reader to the (improved) Excel spreadsheet in the supplementary data - which I assume will be available to readers. The data set could become valuable to future workers.
Three typos, in abstract 'in larger than smaller'
Line 69/70 I had a question mark after 'increase in aggregation' but do not have the manuscript available to the moment.
Line 136 'sill low'.